# Topological and Structural Plasticity of the Single Ig Fold and the Double Ig Fold Present in CD19

**DOI:** 10.3390/biom11091290

**Published:** 2021-08-30

**Authors:** Philippe Youkharibache

**Affiliations:** Cancer Data Science Laboratory, National Cancer Institute, NIH, Bethesda, MD 20814, USA; philippe.youkharibache@nih.gov

**Keywords:** Ig fold, Ig domains, molecular evolution, protein structure, symmetry

## Abstract

The Ig fold has had a remarkable success in vertebrate evolution, with a presence in over 2% of human genes. The Ig fold is not just the elementary structural domain of antibodies and TCRs, it is also at the heart of a staggering 30% of immunologic cell surface receptors, making it a major orchestrator of cell–cell interactions. While BCRs, TCRs, and numerous Ig-based cell surface receptors form homo- or heterodimers on the same cell surface (in cis), many of them interface as ligand-receptors (checkpoints) on interacting cells (in trans) through their Ig domains. New Ig-Ig interfaces are still being discovered between Ig-based cell surface receptors, even in well-known families such as B7. What is largely ignored, however, is that the Ig fold itself is pseudosymmetric, a property that makes the Ig domain a versatile self-associative 3D structure and may, in part, explain its success in evolution, especially through its ability to bind in cis or in trans in the context of cell surface receptor–ligand interactions. In this paper, we review the Ig domains’ tertiary and quaternary pseudosymmetries, with particular attention to the newly identified double Ig fold in the solved CD19 molecular structure to highlight the underlying fundamental folding elements of Ig domains, i.e., Ig protodomains. This pseudosymmetric property of Ig domains gives us a decoding frame of reference to understand the fold, relate all Ig domain forms, single or double, and suggest new protein engineering avenues.

## 1. Introduction

### 1.1. Tertiary Pseudosymmetry of the Ig Fold

We previously established that ca. 20% of known protein folds/domains are pseudosymmetric [1], and that in each structural class [2], the most diversified fold exhibits pseudosymmetry, suggesting a link between symmetry and evolution. Two classes of folds show a higher proportion of pseudosymmetric domains: membrane proteins, with, for example, GPCRs [3], and beta folds, chief among them the Ig fold [4]. The Ig fold is present in over 2% of human genes in the human genome [5] and it is overly represented in the surfaceome/immunome [6,7]. Beyond antibodies, B-cell, and T-cell receptors and coreceptors, the Ig domain is present in a very large number of T-cell costimulatory and coinhibitory checkpoints that regulate adaptive immunity with, in particular, the CD28 family of receptors containing the well-known CTLA-4 and PD-1 receptors and their ligands from the B-7 family [8,9,10]. Overall, the Ig fold accounts for a staggering 30% of cell surface receptors’ extracellular domains [7], making it a major orchestrator of cell–cell interactions. What is especially remarkable with Ig domains is their ability to interact, i.e., self-associate, in both cis and trans trough cell surface receptor–receptor or receptor–ligand interactions. The very notion of cell surface receptor vs. ligand is arbitrary as Ig domains are at the heart of a very elaborate network regulating immune responses through Ig-Ig interactions in cis and in trans [11,12,13,14,15,16,17,18]. A reason for self-interaction in cis or trans lies in its very structure: the Ig fold is pseudosymmetric (Figure 1). While quaternary symmetry of Ig-domain-based complexes is well known, the Ig tertiary structure pseudosymmetry is largely ignored, and we will review this property in terms of both single Ig domains and the recently solved CD19 structure with a novel double Ig fold, a remarkable pseudosymmetrical protein architecture.

### 1.2. Pseudosymmetry and Ancient Evolution of the Ig Fold

A pseudosymmetric domain is formed of two or more protodomains, according to an accepted duplication–fusion mechanism [19], and multiple examples of highly diversified structural folds have been known for a long time [4]. Structurally, it is important to realize that the knowledge of protodomains and symmetry operators define a pseudosymmetric protein domain entirely, apart from a variable linker region, as most often short, chaining protodomains within a domain [4]. Interestingly, a pseudosymmetric domain is a tertiary structure that can also be considered a pseudoquaternary structure and can be analyzed according to symmetry groups (C2 or higher C3, C4…D2, …Dn), with a C2 symmetry prevalence in the known structural protein universe (PDB symmetry statistics: https://www.rcsb.org/search/browse/symmetry, accessed on 27 August 2021). The symmetric assembly of (chained) protodomains is a general property found in biopolymers beyond proteins, as in RNA riboswitches for example [4,20], demonstrating a general duplication mechanism in biopolymers with symmetric self-assembly of the duplicated parts, beyond the realm of proteins, and already present in the RNA world when considering the ribosomal peptidyl transferase center (PTC) [21] as a remnant of an ancient proto-ribosome.

Remarkably, thinking of an ancient peptide/protein world, a very early investigation of immunoglobulins in sequence space [22] suggested that “Each part of the immunoglobulin chains (Fd, Fc, v, c) derives from the repetition of a smaller ancestral sequence (of approximately 20 amino acids)… The following scheme of evolution is suggested: In the first steps, we shall speak only of peptides and not of genes, since it is not known if ancestral peptides were coded by nucleotide sequences. The process started with a primordial peptide which, by a special mechanism, doubled its length and became (pseudo)symmetrical”. One can only be impressed by such an early and insightful analysis, since we are now able to discern pseudosymmetry in Ig domains structurally and delineate protodomains. Further efforts tried to systematize sequence-based pseudosymmetrical analysis of Ig domains [23], but when comparing to 3D structure-based conservation of protodomains, sequence alignment matches do not exactly coincide with structure alignment matches. This is typical of sequence–structure internal symmetry detection [1,3,4], and if sequence matches are real, they imply a shift in the relative position of the sequence in the structure, the evidence of which is difficult to demonstrate. This is an interesting observation nonetheless, albeit anecdotal.

Another early study suggested some “evidence for an ancestral immunoglobulin gene coding for half a domain” [24] based on sequence considerations, postulating that the half domains would correspond to strands AB-CC’ and DE-FG and assemble as an IgV domain, using an accepted nomenclature of Ig domain strands [25,26]. Later, this delineation was supported on structural considerations by McLachlan [27]. These elements are not well known but have been reviewed in the literature (Williams and Barclay 1988) and are supported by our observations based on structural pseudosymmetry of single Ig domains [4]. They are also supported by an observed swap of the GFCC’ protodomain between IgV domains in a dimeric form of CD2 [28,29] and, as we shall see in the following in the case of CD19, the AB-CC’ and DE-FG protodomains interdigitate to form a novel double Ig fold [30].

### 1.3. Converging vs. Diverging Evolution

For many Ig domains, and even more so when also considering structurally similar (Ig-like) domains, such as FN3 or Cadherins, the sempiternal question comes back: are these domains evolutionarily related? Do these domains have a common ancestor? For many highly diversified superfold domains [1,3,4,31], the question is relevant. It is especially relevant as it relates to the various topological forms of Ig domains. The protodomain (half-domain) hypothesis tends to support the IgV domain as the ancestral form in a divergent evolution scenario. Arguments on the origin of the Ig domain have been partially reviewed in the literature [26,32]. In the following, we analyze the structural and topological domains in light of the Ig fold pseudosymmetry and a possible parallel evolution of these topological domains. It is important to note, however, that although structural pseudosymmetry of Ig domains raises the issue of a possible evolution through duplication of an ancestral protodomain, the sequence identity between pseudosymmetric substructures (protodomains) within an Ig domain is in the twilight zone (see Figure 2). Therefore, inferring a possible duplication event from structural symmetry is speculative in the absence of a conserved sequence signature across protodomains. Hence, there is room for interpretation as either divergent or convergent evolution. However, regardless of the ancestral evolution of the domain itself, the observed structural symmetry gives a structural analysis framework, opens the door to graphical representations integrating symmetry, and can be used in the design of antibody fragments down to nanobodies.

### 1.4. Quaternary Pseudosymmetry of Ig Domain Assemblies: Ig Domain Dimerization

Let us briefly review the pseudosymmetric assembly of Ig domains. Twofold (C2) quaternary symmetry has been demonstrated in the very first structural studies on Ig domain dimers, especially of IgV domains [33,34,35]. The IgV-IgV dimerization interface using the GFCC’ sheet was observed in the first Fab structure solved (PDBid: 7FAB) (https://www.ncbi.nlm.nih.gov/Structure/icn3d/full.html?&mmdbid=7fab&bu=1&showanno=1&source=full-feature, accessed on 27 August 2021) [36] and can be considered canonical. It is also found in homodimers such as CD8aa (PDBid: 1CD8) (https://structure.ncbi.nlm.nih.gov/icn3d/share.html?JcP3sd1gGfqXEBEM8, accessed on 27 August 2021) [37], and was in fact already observed in the Bence Jones protein, the very first structure deposited in the PDB (1REI) (https://www.ncbi.nlm.nih.gov/Structure/icn3d/full.html?&mmdbid=1rei&bu=1&showanno=1&source=full-feature, accessed on 27 August 2021) [34], forming a VL-VL interface through that same GFCC’ sheet. In antibodies and antibody fragments such as Fabs or scFvs, pseudosymmetric interfaces are formed by a VH-VL domain pairing. Constant domains such as CH-CL, on the other hand, use the opposite sheet ABED of the Ig domain to dimerize. In some cases, VH domains can also pair through that ABED sheet, as in Fab-dimerized glycan-reactive antibodies [38].

Multiple different dimeric interfaces have been characterized between IgV domains at the N terminus of cell surface receptors and ligands, where a majority of them interact through their GFCC’ sheets, but in a wide variety of orientations. Other pseudosymmetric parallel dimer interfaces for CTLA-4 or CD28 homodimers on T-cell surfaces, for example, use instead their A’G strands and their hinges beyond their G strand to dimerize, and in doing so free the GFCC’ sheet of both IgV domains to interact with other IgV domains in either cis or trans; in particular, in trans using the GFCC’ sheet of their ligands CD80 or CD86, that themselves homodimerize in cis using their strands C”D (see Figure 3), and/or in cis as in the case of CD80/PD-1. It is beyond the scope of this paper to review all possible interfaces, but let us note, nonetheless, that the canonical GFCC’ interface can be considered, in most cases, a “cis” interface, as the domains are parallel, positioning both domains’ C termini towards a given cell membrane, while “trans” IgV-IgV interfaces also using the GFCC’ sheet can be formed from opposite cell membranes in antiparallel, for example between the CTLA-4 (or CD28) receptor on the surface of T-cells with their natural ligands CD80 or CD86 (https://structure.ncbi.nlm.nih.gov/icn3d/share.html?ScHMZf2B5i1xQ3JZA, accessed on 27 August 2021) [39] on the surface of targeted cells (see Figure 3). This is not a rule, however, due to the flexibility of some surface receptors’ hinges, as in the case of PD-1|PD-L1 or PD1|PD-L2 interactions, where IgV domains form a parallel interface while on opposite cell surfaces, resembling the canonical interface and implying an inversion of the PD-1 domain in binding a PD-L domains [40,41,42]. In all IgV interfaces, a quaternary C2 axis of pseudosymmetry is observed, but that axis varies with the relative orientation of IgV dimers. The characterization of the full set of interfaces formed by Ig domains still awaits a review, but let us note that new Ig-Ig binding interfaces are still being discovered, even in those that have been best studied within the Ig superfamily, such as the B7-family with CD86 or CD80 homodimers in cis, PD-L1/CD80 heterodimers in cis using the same interface, CTLA-4/CD80 in trans, and PD-L1/PD-1 in trans using, once again, the same interface [43].

To help with the understanding of Ig-Ig interfaces, for some canonical homodimers, such as CD8aa or VL-VL Bence Jones proteins, the tertiary C2 axis of symmetry combines with the quaternary C2 to lead to a quasi D2 symmetry group. This holds true even for VHVL or CD8ab heterodimers. Interestingly, some inverted homodimer interfaces are observed, where IgV domains are flipped (180 degrees) with regard to each other, resulting in a quaternary axis of symmetry rotated by 90 degrees, still orthogonal, however, to the tertiary C2 symmetry axis and an overall quasi D2 symmetry in considering the quaternary structure (see Figure 3). These structures are closer to idealized dimers and can help us understand a possible origin of the ubiquitous cis vs trans pairing of IgV domains, with parallel vs. antiparallel dimer interfaces. Using a membrane protein nomenclature, the latter is equivalent to an inverted (albeit quaternary) topology as opposed to a parallel one for canonical interfaces. In a canonical VH-VL antibody interface, the CDR loops are on the same side, while in an inverted topology they would be on opposite sides. In Figure 3, we show an idealized representation, comparing both dimer orientations and pseudosymmetries. Even in homodimers, IgV domains can not only adopt a parallel or inverted (flipped) orientation, but they can also adopt tilted orientations in between [44], all related to each other through a rotation around the tertiary C2 axis of symmetry. Further divergence from an idealized form of D2 symmetry, especially in receptor–ligand interactions, shifts these tertiary C2 symmetry axes away from collinearity. It is tempting to postulate that the tertiary pseudosymmetry of the Ig domain may be the structural basis of its self-association in either cis or trans, and for the latter in an inverted (quaternary) topology, yet not excluding a parallel one, as noticed before with PD-1/PD-L1. This is visualized in Figure 3 and beyond through the coloring scheme we have chosen to highlight symmetry with protodomain equivalence. The 3D pictures are either from Chimera [45] or iCn3D [46]. All 3D interactive graphics hyperlinks in the main text of figure legends use iCn3D and can be used directly by readers, not only for visualization but also for further analysis.

In the following, we focus on the tertiary symmetries across all topological variants of Ig domains and dissect the formation of a new tertiary fold—the double Ig domain observed in CD19, which confirms the protodomain hypothesis.

## 2. Results

### 2.1. Protodomain Evidence in Single Ig Domains

As mentioned in the introduction, nature may have used a protodomain duplication–fusion mechanism in a distant evolutionary past to build current day pseudosymmetric domains, and we can gain insight in domain creation from an analysis of tertiary pseudosymmetry of structurally characterized domains to look at domain creation and evolution in terms of their constituting parts. The immunoglobulin fold is at the heart of a very large number of cell surface proteins of the immune system, beyond immunoglobulins themselves, and we have seen that the Ig fold exhibits tertiary symmetry as well as quaternary symmetry, as in CD8 or VL-VL homodimers or the very well-known antibody variable domain association—VH-VL.

### 2.2. The Single Ig Fold Pseudosymmetry and Protodomains Hypothesis

In Figure 4, we show schematically the topology of Ig domains in the variable form (IgV), the shark variable form (VNAR), and the C1 constant domain (IgC) (see more topological variants in Figure 2). An Ig domain can be considered formed as a covalently linked dimer of two protodomains AB-CC’ and DE-FG using the standard strand terminology of immunoglobulin domains [25,26]. Protodomains forming an Ig domain usually align within a 1 to 2 angstrom RMSD range (Figure 2). The two protodomains AB-CC’ and DE-FG are composed of two hairpin domains, AB and CC’, in the first position and DE and FG in the second, linked through BC and EF loops, respectively, and whose relative orientation can be compared to an “inverted topology” (using a membrane protein nomenclature). We will refer to them with these denominations throughout the paper. Naturally Ig domains show more complexity with additional strands along these core central strands to form, in the IgV, for example, ABED and A’|GFCCC”’ sheets with the additional strands, as we shall see. In figures, we use a protodomain spectrum coloring scheme: blue–green–yellow–orange for protodomain strands ABCC’ or DEFG and red for C” (see Figure 1).

One can reconcile all Ig forms (IgV, IgC, …) by considering an Ig domain formation through a pseudosymmetrical combination of protodomains with a variety of linking sequences that can form or remove strands on one sheet or the other (see Figure 2 and Figure 4). This opens the door to the interpretation of the formation of the diverse Ig topologies as a parallel evolution process, leading to Ig domains with variable topologies. A divergent evolution scheme would involve one domain formation with insertion or deletion of sequence elements between strands C and E, leading to all topologies involving strands C’, C”, and D. This is similar to what we have described previously in pseudosymmetric polytopic membrane protein formation [3].

### 2.3. Protodomain Evidence in the CD19 Double Ig Fold

#### 2.3.1. Characterization of the CD19 Ig Domains in Sequences and Structural Databases

Recently the structure of CD19 has been solved [30]. Prior to that knowledge, any textbook and sequence database would have presented CD19 as formed of two Ig domains of C2 topology in tandem, and, in fact, this is still the case today in some databases, where UNIPROT (P15391) and other databases “confidently predict” CD19 extracellular region as composed of two Ig domains belonging to the IgC2 set (see discussion hereafter). From a sequence perspective, they are not that far off, but the devil is in the details and sequence-based Ig domains are just not folding as two single Ig domains in tandem, but as one double Ig domain. Could a sequence analysis reveal a double Ig domain? Probably not, sequence matching algorithms clearly detect two Ig domains, however a detailed analysis shows that there are two areas unique to the CD19 extracellular domain sequence, first the two “Ig domains” are not connected in tandem but separated by a long linker, that we now know is structured as a small (inverter) domain (see Figure 5), and also a short “protodomain linker” within each of the “detected Ig domains”. This is difficult to interpret for sequence matching algorithms using defined topological patterns (I-set, V-set, C1-set, and C2-set). What is more surprising, however, is that an automated structural pattern-matching algorithm (ECOD) would assign the double Ig domain as a V-Set followed by an I-set [47].

#### 2.3.2. Protodomain Evidence in the CD19 Double Ig Fold

The double Ig fold adopted by CD19 [30] can be understood in terms of protodomains. In regular IgVs, AB-CC’ (p1) and DE-FG (p2) protodomains are assembled in antiparallel with an internal C2 pseudosymmetry, through the characteristic {CDR2 loop + C” strand + C”D Loop} structured “linker” (see Figure 5). Shorter topologies (I-set, C1-set, and C2-set) differ in the linker between protodomains (see Discussion below). It is easy to misidentify the strands C’ or D in the case of CD19 and to call for IgC2 domains. In CD19, strands C’ and D are observed and participate in consecutive protodomains AB-CC’ (p1) and DE-FG (p2) that are separated by a very short linker that forces them to remain parallel, unlike any known Ig domain topology. This implies a structural resilience of protodomains as they can associate in parallel or invert. To reconcile these elements with the sequence analysis “the first Ig domain detected” adopts a parallel topology, and so does the second, something novel. Both “Ig domains of parallel protodomain topology”. as identified by sequence analysis, can then assemble through a long linker that allows them to assemble/intertwine in an inverted topology (antiparallel with a C2 symmetry), juxtaposing their two BED sheets together and the two A|GFCC’ together as two long beta sheets facing each other (see Figure 5).

CD19 offers a higher complexity buildup than single Ig domains in combining four Ig protodomains to form an interdigitated pseudosymmetric double Ig domain, a fold never observed before (Figure 5D). Crystallographers describe the structure they solved as a domain swap [30], involving protodomains. CD19 topological innovation, however, results from a much more subtle and interesting folding. A case of protodomain swap in CD2 is known (https://structure.ncbi.nlm.nih.gov/icn3d/share.html?AdsvRptE9BHqWnN56, accessed on 27 August 2021) and it is not the same as a CD19 fold (Figure 5). Noticeably, the authors at the time had hypothesized a possible protodomain duplication in the early evolution of Ig domains [28], i.e., the half-domain hypothesis described earlier [24,27], and even engineered higher order oligomers swapping protodomains [29]. What nature has achieved, however, in the case of CD19, is a true folding innovation, leading to a novel double Ig topology. We can analyze the double Ig domain folding of CD19 (https://structure.ncbi.nlm.nih.gov/icn3d/share.html?LGcQe5UM4nFx7dnL8, accessed on 27 August 2021) as formed by two Ig domains of “parallel topology”, where sequential protodomains AB-CC’ (p1/p3) and DE-FG (p2/p4) are linked by short linkers (p1-p2 and p3-p4), the two parallel domains assemble pseudosymmetrically with an inverted topology (using a membrane protein denomination) (see Figure 5). Amazingly, in doing so, it also forms two composite Ig domains, with protodomains p1 and p4 related by a local C2 symmetry, and a similar occurrence takes place for p2 and p3. It represents a marvel of topological engineering and folding. It is worth noting that from a sequence standpoint the Ig domain formed of p2 + p3 has an inverse Ig topology, swapping sheets BED (p3) and GFCC’ (p2), as p2 precedes p3 in sequence. In essence, p2 and p3 are swapped and form an inverse Ig domain (see Figure 6). This is equivalent to a circular permutation of protodomains yet obtained purely from folding. Ultimately, the CD19 double Ig domain forms two fused long sheets of: (BED)1(DEB)2 vs. (A’|GFCC’)1(C’CFG|A)2 (Figure 5D). The A’|GFCC’ sheets are fused through their C’ strands and their opposite BED sheets are fused through their D strands. A canonical IgV dimer or a swapped dimer would have GFCC’ sheets, facing each other with nonbonded interactions rather than through lateral hydrogen bonded beta strands (see Figure 4F,G and Figure 6). This is truly a remarkable structural and topological innovation. More details on residue interactions between CD19 protodomains can be found in Appendix A.

## 3. Discussion

### 3.1. Structural and Topological Plasticity of Ig Domains

The accepted nomenclature on beta strands [25] has progressively been used to develop a classification of Ig domains, distinguishing four main classes of Ig domain topologies: V-set, C1-set, C2-set, I-set. All Ig domains in these classes share only four central beta-strands B,C,E,F, forming two loops straddling the two sheets of the barrel BC (CDR1) and EF, and a fifth G strand at the C terminus, forming the FG loop, called complementary determining regions 3 (CDR3) in antibodies (we use the CDR nomenclature of antibodies here for all IgV domains). Two to five lateral strands of A/A’ at the N terminus and C’, C”, D, which we will describe below, define and distinguish these four classes [25,48]. Today, many topologies and associated families have been classified and organized in a hierarchical tree (https://www.ncbi.nlm.nih.gov/Structure/cdd/cddsrv.cgi?uid=cl11960, accessed on 27 August 2021) [49]. Some authors have proposed additional sets or subtypes (C3, C4, V, H, and FN3) [50], but these complexify the topological landscape, and the more diverse the classification becomes to account for small details, the less we can see the commonalities and a possible common evolutionary scheme that led to today’s diversity in topologies. The number of all possible topological variants accounting for specific details of Ig domains in each and every functional family is quite high when considering that lateral strands can be topologically present or not, and can split and swap between the two sheets of the Ig sandwich barrel, giving rise to a large combinatorial structural ensemble. Adding sequence diversity, one can understand the evolutionary success and diversification of these domains. Adding multiple Ig domain chaining (tandemization) and chain oligomerization adds even more complexity and diversity.

In this paper, we take a step back and look at the common symmetric architecture of all Ig domains: a protodomain decomposition allows us to reconcile all Ig domain topologies in light of that common pseudosymmetric domain architecture, which could imply either a variable fusion mechanism linking these protodomains together to produce various domain topologies (parallel evolution) or a divergent evolution from symmetrically fused protodomains. CD19 escapes the current classification and exemplifies a new topological double Ig domain innovation that can be explained by how Ig protodomains fold and combine. At the same time, the protodomain decomposition validates the single Ig pseudosymmetric domain organization.

### 3.2. Ig Domain Classes and Their Topologies

All Ig domains can be related, however protodomains within Ig domains of different topologies exhibit symmetry breaking when considering AB-CC’ and DE-FG protodomains, as some of the lateral strands A/A’, C’ or D are not present. Let us first go over these topological variations:

In IgVs, the A strand splits in A/A’ (usually through a proline or a set of glycines), with A extending the A|BED sheet in antiparallel while A’ snaps to extend the A’|GFCC’|C” sheet in parallel and straddles one lateral side on the sandwich. In some other IgVs, such as in CD4, the A strand is reduced to A’ to participate in that sheet only. The IgV is the only Ig domain with a C” strand that extends the A’|GFCC’|C” sheet and forms in doing so a CDR2 loop (C’-C”). However, the C” strand can also participate, instead, to the ABED|C”. This is the case of CTLA-4 [39], which has a split A/A’ on one side of the sandwich and a C” on the other side, and therefore closes the two open sides of the Ig sandwich to resemble a more a fully connected (H-bonded) 10-stranded quasi closed barrel ABEDC”C’CFGA’. In doing so, it also blocks possible lateral strand–strand (backbone level) quaternary interactions, while offering, overall, more surface for non-bonded quaternary interactions all around the barrel.

In IgC1s, the A strand lies entirely on the A|BED sheet, there is no split A/A’ with A’ snapping to the other sheet, which is reduced to GFC (no C’).

In IgC2s, the A strand lies entirely on the A|BE sheet (no D). There is no split A/A’ and a GFCC’ sheet.

In the I-set, as in many IgVs, A and A’ are split and A participates in the A|BED sheet in antiparallel, while A’ swaps to the opposing sheet in parallel to strand G: A’|GFCC’. The C’ strand is very short (2–3 residues), followed by a very straight C’D linker to the D strand on the opposite sheet.

In VNAR domains, the heavy chain-only variable domain of sharks [51,52], the topology seems to be lying between the I-set and the V-set if one considers the similarity between the protodomain linkers, i.e., with the so-called HV2 region instead of the CDR2/C” region in IgVs, which can be considered to be composed of a very short C’ before the C’D linker straddling the D strand. The topological organization is similar to the I-set (Figure 2) and the C’D linker that even aligns structurally to C’D is usually considered part of the HV2 variable region in e VNARs. It is tempting to see an evolutionary relationship.

In cadherins, the A strand lies entirely on the A’|GFC sheet (no C’) facing a BED sheet.

In FN3, similarly does not split, the A strand but lies entirely on the A|BE sheet (no D strand) vs. the GFC|C’ sheet.

In CD19, for each of the two fused structural Ig domains, the A’ strand lies entirely on A’|GFCC’ sheets and BED sheets are formed opposite to them. Both Ig domains extend each other’s sheets in a unique way, as two fused long sheets: (A’|GFCC’)1(C’CFG|A)2 vs. (BED)1(DEB)2, which is only possible through an inversion of one domain through a unique folding/swapping mechanism (Figure 5D).

There are many papers reviewing differences between topological variants of the Ig domain [53,54] and a few hypotheses have been proposed on a divergent evolution scenario between them, with the IgC2 as the oldest one [54]. The I-set [55,56] is intriguing, as it is sometimes described as a V-set truncated on one side, missing the C′′ strand. The I-set is topologically and structurally similar to the VNAR domain, including on strands A/A’B-CC’ and DE-FG, with a very short C’ and, more surprisingly, with a matching C’-D linker, missing the CDR2 loop of IgVs. VNAR, however, is truly a variable domain with V(D)J recombination [52,57]. The four main classes of Ig domains, as well as VNAR and the FN3 domains, are shown in Figure 2 as topology/sequence maps that visualize topological/sequence alignments based on structure.

Multiple cell surface receptors have an IgV-like domain at the N-terminus for ligand-binding, and many of their ligands also have an IgV-like domain at their N terminus. Ig-C1 or Ig-C2 domains often follow N-termini IgV-like domains in tandem. This is the case of CD4 [58], which shows a tandem fused IgV-IgC2 (the G strand of the IgV and the A strand of the IgC2 are fused to form a rather rigid two-domain tertiary structure, that itself is tandem, being duplicated as four extracellular domains (IgV-IgC2)2. CD2 [59] also presents a tandem IgV-IgC2, yet a hinge separates the IgV G strand from the IgC2 A strand, giving the variable domain more flexibility for dimerization of the IgV domain (see swapped domain in Figure 5) and ligand interactions. All these topological variants produce sheets and side strand exposures that can lead to a variety of interaction and Ig-Ig dimer interfaces. It is beyond the scope of the present paper to review all possible interfaces, there are so many, except as it relates to canonical and inverted IgV domains and CD19 double Ig domains (see Figure 5). 

### 3.3. Protodomain Symmetry-Breaking Elements

#### 3.3.1. The Protodomain Linker Region Varies among Ig Domains

This whole C[c’c”d]E region distinguishes the various Ig domains. They can all be reconciled by understanding the linking of two protodomains: AB-C[C’–linker–D]E-FG, forming (or removing) strands, partially breaking the symmetry and leading to topological variants. For example, comparing the two protodomains in an I-set domain through self-alignment (https://structure.ncbi.nlm.nih.gov/icn3d/share.html?tBzknU8spazpwCup7, accessed on 27 August 2021) where the C’ strand that would match the G strand by symmetry in an IgV (shown in orange in Figure 2) morphs into a straddling linker between the two strand sheets, between strands C and D. 

In FN3 domains [4,60] the D strand is missing, leading to a symmetric match only for strands B<>E, C<>F, and C’<>G with a C’E linker (Figure 2). Similarly, In the structurally related cadherin domains [61] the C’ strand is missing, leading to a symmetric match of A<>D, B<>E, and C<>F, with a CD linker. The Ig domain’s topological variants can be compared through 2D topology/sequence maps and structure-based sequence alignments in Figure 2. The fusion of two protogenes and evolution would determine the length and secondary structure of protodomain linkers, and ultimately the function of the resulting Ig domain.

We cannot objectively reconstitute an old evolutionary process that gave rise to Ig domains, but if we analyze multiple pseudosymmetric folds, we can infer that a substructure can duplicate with variable linker regions, these regions can be short or long and enable substructures to self-assemble pseudosymmetrically, breaking symmetry in a variety of ways. Naturally, a divergent evolution scenario after duplication/fusion through insertions and deletions would also result in a variety of topologies. The two evolutive scenarios are certainly not exclusive and, in fact, are likely when considering sequence similarities such as the CCW(L) signature in Ig domains [4]. In the case of a simple duplication event leading to a C2 pseudosymmetric structure, as in membrane proteins, they can assemble in antiparallel to lead to a so-called inverted topology, or in parallel, as in the case of CD19. Depending on the number of secondary elements in a duplicated substructure, a short or long linker may be needed to achieve a parallel or an inverted topology. A single Ig domain, when considering protodomains made of four strands (two hairpins), would need a rather long linker to achieve an inverted topology of two consecutive protodomains. This is the case of an IgV domain where that linker can form a secondary structure, with a CDR2 loop between a C’ and a C” strand that complements the sheet, and then a turn C”D linking across the domain to the ABED sheet. If, however, the linker is short or non-existent, what are the choices to achieve an antiparallel protodomain organization? Sacrifice either the C’ or D strand, using the sequence stretch to link the two sheets obtaining an IgC or an I-set (no C’) or, alternatively, an IgC2 or an FN3 domain. On the other hand, in the case of CD19, a short linker between the A’B-CC’ (p1) and DE-FG (p2) protodomains enables a parallel organization, which can then assemble in antiparallel with a duplicated copy (p3 + p4), thanks to a small inverter domain linker placed between the two “domains” (see Figure 5).

#### 3.3.2. The Common Core

The immutable architectural element, the common core of Ig and Ig-like domains, is the BC-EF intertwined symmetric straddle (Figure 1), which is at the heart of not only Ig fold SCOP b.1, but also other folds such as SCOP-folds b.2 (p53-like transcription factors) or as b.3 (prealbumin-like) [2], which are all classified together as immunoglobulin-like in other classifications (ECOD, CATH) [62,63]. This symmetric straddle that encompasses BC (CDR1) and EF loops in symmetrically related positions is a signature of the Ig-like “inverted” topology, where strands B-C and E-F substructures are intertwined in antiparallel.

#### 3.3.3. The N Terminus Region: Strand(s) A-A’ Can Also Vary, Resulting in a Partial Symmetry Breaking

In Ig domains, the A strand is quite interesting and distinguishes various Ig domains. The A strand in an IgC1 will form a regular hairpin and strands AB of protodomain 1 will pair with DE of protodomain 2, but in the context of an IgV, the A strand will split in two, A and A’, usually through a proline or glycines, and the A’ will participate in the opposite sheet in parallel. In some cases, the A strand lies only on the opposing sheet as A’, and the protodomains will remain pseudosymmetric only on three strands, BED vs. CFG. 

In summary, when considering Ig domains and comparing the various topologies, independently of the CDRs in antibodies, BCRs, and TCRs, the most variable regions, structurally and topologically, are on the edges of the domain (barrel sandwich), based on how the protodomains are linked, leading to multiple topologies between C and E strands, with, in particular, the C’-CDR2-C”-C”D structured linker in IgVs, as well as the A strand topology variants at the N terminus, with strands A (ABED sheet), or A’ or split A-A’(A’GFCC’ sheet), participating in both sheets (see Figure 2). Our approach sees all topological variants of the Ig domain through the lens of the pseudosymmetric assembly of protodomains, tied together through a linker of variable length, and a secondary structure. With the knowledge of the double Ig fold, we have been able to rationalize protodomain assembly with that same lens, yet with a higher complexity, and we have highlighted two forms of a sequential Ig domains: with consecutive protodomains folding in parallel vs antiparallel (inverted), depending on the inter-protodomain linker’s length. CD19 folding is the only instance (so far) of a double Ig fold in the structural database. This fold is unique in more than one way—in folding as parallel Ig domains and in the pseudosymmetric assembly of these parallel Ig domains, resulting in composite structural Ig domains, with an instance of an inverse Ig fold, resulting from a complex protodomain swap. It is important to note that it is not a simple exchange of secondary structure elements between domains, referred to usually as “domain swap” [30,64,65]. The Ig domain composed of structurally swapped protodomains p2 + p3 is an inverse Ig domain, equivalent to a circular permutation in sequence. 

#### 3.3.4. Pseudosymmetry across Ig Domain Variants

Figure 2 shows structure-based sequence alignments of protodomains for a variety of Ig domain types I-set, V-set, C1-set, C2-set, VNAR, and FN3 and quantifies the pseudosymmetry of Ig topological variants using the RMSD (root mean square difference) as a measure of structural similarity. Ig variants’ protodomains superimpose between 1 and 2.5A RMS (C1: 1.19 A, FN3: 1.8 A, C2: 1.98 A, I: 2.46 A and IgVs 1.6–1.8 A, and VNAR 2.31), similarly to domain-level structural homologies. RMSD was lower for IgC1 and IgV protodomains than for I-sets’ or VNAR’s in the selected examples, but it would require a systematic survey to give significance to these differences in structural similarity across topological variants. The RMSD is a broad measure and substructural elements can be superimposed with a very low RMSD, while others show a higher variability. Protodomains differences become larger (in overall RMSD) when moving to Ig-like proteins, such as cadherins, for which it has been argued that they are not related to evolution [61], while FN3 domains behave very much like Ig domains in terms of pseudosymmetry matches, and can even present higher sequence similarity (Figure 2B).

### 3.4. Searching for CD19 Homologs in Sequence or Structure and for Inverse Ig Domains

We performed a sequence search for a hypothetical inverse Ig domain based on a composite permuted sequence of protodomains p2–p3, but did not identify any inverse sequence, except in CD19 orthologs. If we search structurally [66], we get over 7000 hits, however, due to the domain pseudosymmetry, which matches a regular Ig domain with permuted protodomains with a very low sequence homology, only 10 hits had a sequence match of above 20% identity. If we examine matches above that threshold, they also appear to be Ig domains with regular topology. The very top hit, exhibiting 23% sequence homology, is a structural Ig domain of the Togavirin double Ig domain in the matrix remodeling associated protein 8 MRXA8 (PDBid 6JO8 and homologs). It presents what looks like a circular permutation of only half the protodomain, consisting of the AB strands. However, it is not a permutation at the gene level, but a domain swap of the AB substructure. In sequence, we have two chained Ig domains through an extra strand H as a linker {AB}1|{CC’C”-DE-FG)}1|H|{AB}2|{CC’C”-DE-FG)}2; in structure, however, the two Ig structural domains related by a C2 pseudosymmetry have the swapped topology {CC’C”-DE-FG)}1|H|{AB}2-{AB}1|{CC’C”-DE-FG)}2, as shown in Figure 5, in a head to head swapped “dimer”, yet a tertiary structure [67]. The first structural Ig domain appears as if it were a circular permutant in sequence with strands CC’-C”-DE-FG-H-AB, composed of strands CC’-C”-DE-FG of the first Ig sequential domain and AB of the second, linked through a domain “linker”, forming a new substructure with a new strand H between the first G strand and the second A strand in sequence, (i.e., a GH loop/strand H/HA loop linker). The two composite structural domains use what would be the BC (CDR1) loop as a linker to swap.

#### Interdigitated Double Domains vs. Tandem Domains

While unique among Ig domain topologies, the interdigitated folding of Ig protodomains in CD19 double Ig domain rather in tandem (see Figure 5) is reminiscent of that of the double Tudors. Two sequential Tudor domains can swap their protodomains p1 + p4 and p2 + p3 (PDBid: 2GF7) (https://structure.ncbi.nlm.nih.gov/icn3d/share.html?TgULdqAYtGCK52NS8, accessed on 27 August 2021), while tandem Tudors fold sequentially as p1 + p2 and p3 + p4 (PDBid: 1XNI) (https://structure.ncbi.nlm.nih.gov/icn3d/share.html?SH5xxEaVv5z2Y5Fi6, accessed on 27 August 2021) [68,69] (see Appendix A). This double Tudor folding and the resulting topology observed in the Jumonji domain containing 2A (JMJD2A) and the retinoblastoma-binding protein 1 (RBBP1) proteins has intrigued scientists who solved their structures [68,70], as they note: “It will be extremely interesting to understand the principle underlying the distinct folding of the double tudor domains of JMJD2A and 53BP1 despite their sequence similarity”. In fact, small barrels of SH3 topology such as Tudors exhibit a pseudosymmetric tertiary architecture [69], and their protodomains can interdigitate between consecutive sequential domains in forming pseudosymmetric double domains. The authors concluded: In the case of small barrels with a hairpin-strand protodomain as in Tudors, it will form either tandem domains (Tandem Tudors) (https://structure.ncbi.nlm.nih.gov/icn3d/share.html?kyrVG3gcjmLytS2f8, accessed on 27 August 2021) or a double Tudor domain (https://structure.ncbi.nlm.nih.gov/icn3d/share.html?riGy7LgsVdiPYqX69, accessed on 27 August 2021), but in that case long links are not a geometric requirement to invert protodomains and interdigitation of protodomains may be driven more by their sequence affinity.

In the case of CD19, however, we have proposed the hypothesis of protodomains as folding units and inter-protodomain linker length as a distinctive element allowing Ig domains to fold in parallel (short linker) vs. in antiparallel (long linker); in the latter case, enabling structural tandem domain formation, while in the former, enabling interdigitation. If protodomains form stable supersecondary structures, then the linker length will enable either structural tandem or interdigitated domain formation. Pseudosymmetric assembly of protodomains as a domain requires a linker whose length depends on their folded topology. In the case of Igs with two-hairpin protodomains, a long enough linker is needed to invert the second protodomain vs. the first one as a single Ig, but a very short linker will prohibit two consecutive protodomains from folding as a closed Ig domain and lead to an open parallel Ig domain that can interdigitate with a copy of itself, as in CD19. Linker length is known to control intrachain domain pairing of VH and VL domains as either scFv (single chain Fv) fragments vs. interchain dimeric assembly as diabodies [71,72]. The same principle of pseudosymmetric assembly of domains is observed in the pseudosymmetric protodomain assembly, forming either tandem domains or interdigitated domains.

### 3.5. Orientational and Dynamic Plasticity of IgV Quaternary Interfaces

Many quaternary (dimeric) interfaces of IgV domains involving the GFCC’ sheet interaction adopt a canonical interface, as in antibody variable domains VH-VL or in CD8 homo- or heterodimers (a VH-VL domain pair aligns with a CD8aa dimer within 1.92 A RMSD over 192 residues with 23% sequence identity (https://www.ncbi.nlm.nih.gov/Structure/icn3d/full.html?showalignseq=1&align=7bz5,1cd8&atype=0, accessed on 27 August 2021). Interfaces can also exhibit a slightly rotated, even inverted, interdomain orientation (see earlier). X-ray structures used for our analyses represent static averages on dynamic proteins. We have been performing molecular dynamics simulations on some of these IgV-IgV interfaces at the heart of cell surface receptor–ligands interactions, as well as VL-VH and VL-VL pairs, to understand their dynamic behavior and stability. This is especially important when constructing chained VL-VH antibody fragments in scFv and diabody forms, as mentioned in the previous paragraph, where not just pairing of VH and VL domains, but also their dynamics, can be modified by VL-VH linkers vs. native antibody dimeric forms (unpublished work), likely to impact antibody–antigen interactions. Recently published work in the context of VH-VL interdomain dynamics in antibody fragments using molecular dynamics and NMR showed that these interfaces fluctuate in relative orientation by a few degrees, coupled with conformational rearrangements of CDR loops [73,74], underscoring the importance of the IgV dimer orientational degree of freedom in function. From these studies, it should be anticipated that the dynamics of dimeric IgV-IgV interfaces between receptors and ligands on the surface of cells should play an important role in cell–cell communication and signaling. This is an important topic for follow up studies, and may also be of importance for the design of nanobodies as checkpoint inhibitors to compete with natural ligands binding a cell surface receptor, either alone or in harnessing chimeric antigen receptors (CARs) for CAR T-cell therapies [75].

### 3.6. Pseudosymmetric Residue Interaction Networks

X-ray structures have trained us in comparing averaged structures, and one of the most striking features is the conservation of the VH-VL “canonical” interface across a wide variety of antibody variable domains sequences, especially when compared to the diversity of Ig-Ig domain interfaces in the already known protein universe. Residue interaction networks in heterodimer vs. homodimer interfaces differ, and consequently their interaction surfaces, due to different atom-level contacts for different sequences, even if some specific key interactions are conserved across antibody variable domains. Only homodimers, such as VL-VL dimers, will have the same surface of interaction between equivalent substructures in contact. In canonical IgV-IgV interfaces, there is a rewiring of side chain contacts with different sequences but an overall geometry conservation. If we compare the canonical IgV interface in an antibody VH-VL heterodimer (B38 antibody 7BZ5) vs. a VL-VL homodimer (Bence Jones protein 1REI), their quaternary structures align within 1.53A for 202 residues with 57% sequence identity (https://www.ncbi.nlm.nih.gov/Structure/vastplus/vastplus.cgi?uid=1REI, accessed on 27 August 2021) (see Figure 7). Interacting residues at heterodimeric domain interfaces have co-evolved in conserving quaternary geometries.

#### 3.6.1. Protodomains Interfaces in Single IgV Domain

As for IgV heterodimeric canonical interfaces, we can analyze the Ig domain tertiary fold geometry conservation under the prism of a coevolution of residues at protodomain–protodomain interfaces. In Ig domains, the interface involves the central strands B-C and E-F (we use color to highlight symmetry related strands; contacts B|E are shown in green and C|F in yellow in most figures), with a stringent core interface conservation despite sequence divergence. The central strands are involved in lateral strand–strand interactions (B|E and C|F), leading to beta sheet formation and cross-sheet B-F/C-E core contacts. We can observe a conserved pattern of residue interactions across IgV domains (see Figure 7).

#### 3.6.2. Protodomains Interfaces in CD19 Double Ig Domain

CD19 is formed out of four protodomains, as described earlier (Figure 5 and Figure 6). Their residue interaction networks are presented in Appendix A. Protodomains 1 and 4 exhibit a similar interaction pattern to regular IgV domains, while protodomains 2 and 3 form an inverse pattern. In addition, protodomains 1–3 and 2–4 form a unique pseudosymmetric interface through their C and D strands, with lateral C1-C2 (blue–blue) and D1-D2 (orange) strand H-bond interactions fusing the composite Ig domains through their beta sheets to form the double Ig domain (see Figure 4 and Appendix A).

### 3.7. Pseudosymmetry and Nanobody Design

Nanobodies, i.e., single IgV domains such as VHH domains from camelids, or VNAR domains from sharks, can bind to protein targets using not only their CDRs (with an especially long CDR3 in the case of shark domains [76,77], they can also interact with target IgV domains using their respective GFCC’ framework sheets, in contrast to multidomain antibodies. For example, a nanobody has been shown to target the PD-L1 GFCC’ sheet similarly to the natural PD-1/PD-L1 pairing (https://structure.ncbi.nlm.nih.gov/icn3d/share.html?eCkGRhx16cmbmjQUA, accessed on 27 August 2021) acting as a PD-1 checkpoint inhibitor [78,79]. While the PD-1/PD-L1 complex is C2-pseudosymmetric, it is slightly distorted compared to the canonical interface offering a quasi-D2 symmetry (see Figure 3), shifting the tertiary C2 axes of symmetry of the individual extracellular IgV domains (PD-1 and PD-L1) away from a quasi-collinearity (see Figure 2A and Appendix A). It can be expected that nanobodies could be engineered to pair with target IgV domains using that GFCC’ sheet in a variety of orientations, from canonical dimer interfaces to inverted interfaces, using symmetric pairing as a guide. If we look at IgV-IgV quaternary interfaces in cell surface receptor/ligands beyond PD-1/PD-L1, such as in CD28 and/or B7 receptors, they exhibit a wide variety of interfaces, many, however, use their GFCC’ sheets to interact (see Figure 3), even if they depart from the canonical interface used by VH-VL domains in antibodies or by CD8 dimers. In fact, canonical interfaces offer a surprising conservation of quaternary pseudosymmetric structure despite different interactions between varying residues.

Pseudosymmetry could be used as a guide in engineering nanobodies, especially in leveraging the understanding of framework-based interfaces with target IgV domains, since a nanobody can bind its target through its GF|CC’ sheet, while regular paired antibodies do not have a framework binding option (they use that sheet to pair their variable domains (VH-VL). Leveraging quaternary pseudosymmetric binding can be a route to designing nanobodies by engineering their outward-facing residues. It should also be possible to symmetrize their GF|CC’ sheet (for residues facing outward, keeping today’s core residues that have co-evolved). Although the interest in a pseudosymmetric nanobody may be limited, another way of using symmetry as a design principle would be to, instead, invert the binding direction of nanobodies to interact with IgV targets in either cis or trans when integrated to a cell surface receptor. We plan to test ideas along these lines in the future.

In stand-alone nanobodies, further symmetrization could be envisioned (apart from the asymmetry of CDR2 in VHH or HV2 in VNAR) to lead to bivalence in terms of CDR1/3, since the EF loop is equivalent to CDR1 (BC), and the CC’ loop is equivalent to CDR3 (FG) through symmetry. By interpreting EF and CC’ loops that way, and if CDRs are effectively the determining regions in binding specific targets, a nanobody could also be seen as bispecific. These ideas are speculative at this stage and are only mentioned as conceptual pseudosymmetric designs. Another lesson from the CD19 pseudosymmetric fold is highlighting the folding of an inverse Ig domain that could be engineered by circularly permuting Ig protodomains.

## 4. Methods

Methods to identify pseudosymmetry in protein domains have been published previously [4]. Software programs used in this work were iCn3D [46]; CEsymm [1]; SYMD [80], which has been integrated recently to iCn3D as a web service; and Cn3D for interactive multiple structure alignment [81]. In addition, NCBI structure databases and structure comparison servers (MMDB/VAST/VAST+) [82], the PDB servers [83], DALI server for structure searches [66], BLAST for sequence searches [84], UNIPROT [85], and NCBI CDD annotations [49] for structure- and evolution-related annotations, in particular the latest IgV-set CDD (https://www.ncbi.nlm.nih.gov/Structure/cdd/cddsrv.cgi?hslf=1&uid=409355, accessed on 27 August 2021), IgC1-set CDD (https://www.ncbi.nlm.nih.gov/Structure/cdd/cddsrv.cgi?uid=409354, accessed on 27 August 2021), and many instances of IgC2 CDD (https://www.ncbi.nlm.nih.gov/Structure/cdd/cddsrv.cgi?hslf=1&uid=409491, accessed on 27 August 2021) and IgI CDD (https://www.ncbi.nlm.nih.gov/Structure/cdd/cddsrv.cgi?hslf=1&uid=409545, accessed on 27 August 2021) with detailed Ig sequence–structure features, down to each and every strand of these domains, were used. We used the original PDB numbering for the selected Ig domains in this study. Other commonly used reference numbering for IgV domains, Kabat [86,87] and IMGT [88], are indicated on topology/sequence maps (Appendix A and Appendix A respectively).

Coloring is a very important method to display and highlight properties, and for most figures, as in associated iCn3D links, we used the coloring scheme: blue–green–yellow–orange for protodomain strands ABCC’ or DEFG and red for C”, with grey for loops. For structural alignments, conserved residues are in red and mapped non-conserved residues are in blue. Interactions are colored grey for Van der Walls contacts, green for H-bonds, cyan for salt bridges, red for aromatic pi-stacking, and blue for pi-cation.

## 5. Conclusions

The analysis of pseudosymmetric domains, while not demonstrating a folding pathway, demonstrates stable folded elements that we call protodomains. The Ig domain can be considered to be formed of two protodomains and the recent identification of a double Ig domain present in CD19 gives a confirmation of the protodomain hypothesis and shows the ability of Ig protodomains to fold independently and assemble in either parallel or antiparallel, similarly to membrane proteins of parallel vs inverted topologies. We have also seen the propensity of Ig domains to pair symmetrically in either parallel or inverted “quaternary topologies” as a result of the tertiary pseudosymmetry of Ig domains themselves.

The evolutionary origin of the Ig domains of varying topologies is not entirely solved by this decomposition, yet the decomposition in protodomains involving a symmetric combination, as well as symmetry-breaking elements, offers a number of possible scenarios throughout evolution to explain these topological Ig domain variants. This also tends to suggest that the IgV may be the ancestral topological form of single Ig domains, but the double Ig domain would have had to have its own path.

Importantly, the pseudosymmetric deconstruction of Ig domains provides a frame of reference for innovative molecular design. We have seen the formation of interdigitated protodomains in double domains through the largely unexplored possibilities of linkers to control folding. We have observed an inverse Ig domain within the double Ig domain, and one can extrapolate to possibly engineer (circularly) permuted Ig domains for targeted application. As nanobody applications expand, harnessing the pseudosymmetry of single Ig domains can offer multivalent binding surfaces for cell surface ligand designs and efficient targeting of Ig-based cell surface receptors, to lead to next generation checkpoint inhibitors and chimeric antigen receptors for CAR T-cell therapies in the burgeoning field of immunoengineering.

## Figures and Tables

**Figure 1 biomolecules-11-01290-f001:**
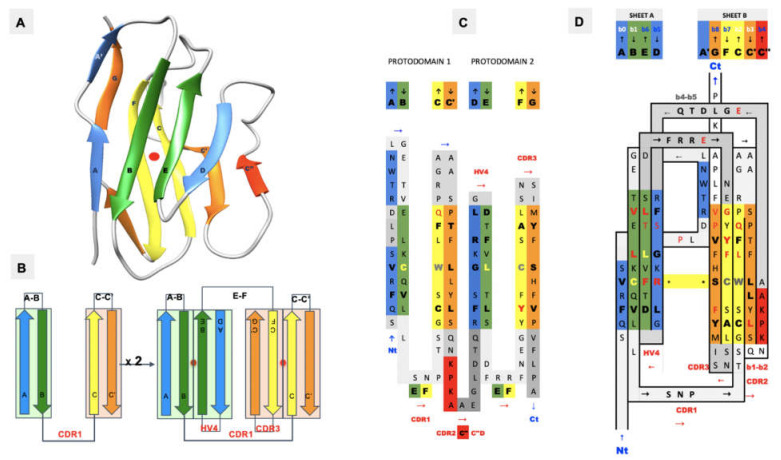
IgV domain deconstruction into pseudosymmetric protodomains with an inverted topology: (**A**) IgV domain—the color scheme blue–green–yellow–orange is associated with each of the individual strands of protodomain 1 A B-C C’ and protodomain 2 D E-F G, which align between 1 and 2A in most IgVs and assemble pseudosymmetrically with a C2 axis of symmetry perpendicular to the paper plane. (**B**) This corresponds to an inverted topology (using a membrane protein nomenclature) between the two protodomains. (**C**) They invert through the linker [CDR2-C” strand-C”D loop]. (**D**) The resulting IgV topology shows the self-complementary assembly of the protodomains through their central strands, the B|E and C|F strands. Symmetry breaking occurs through the C” and A’ strands. In IgVs, unlike IgCs, the A strand splits in two through a proline or a number of glycine residues and participates to the two sheets A B E D and A’|G F C C’|C”. In figure (**A**), we use PDBid 2ATP, where the A’ strand is well formed. In (**D**), the sequence/topology map for the CD8 sequence. iCn3D link for CD8 (https://structure.ncbi.nlm.nih.gov/icn3d/share.html?JcP3sd1gGfqXEBEM8, accessed on 27 August 2021) (PDBid 1CD8) is shown.

**Figure 2 biomolecules-11-01290-f002:**
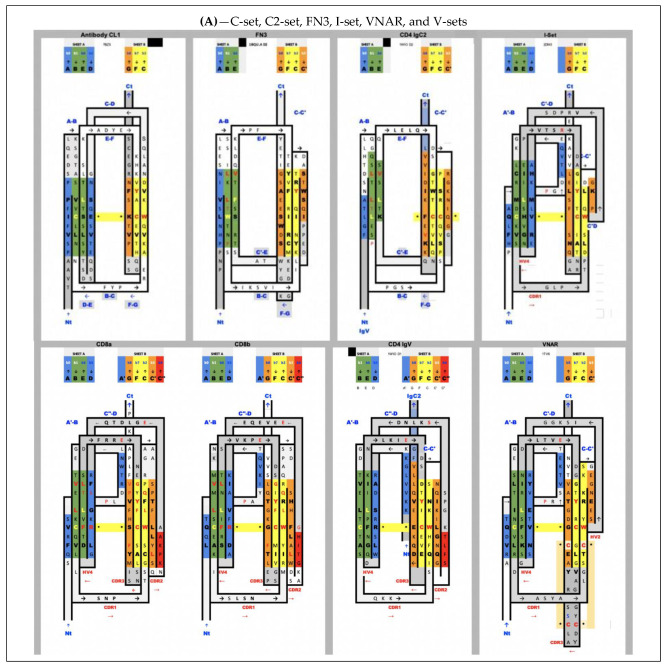
Sequence/topology maps comparison of topological variants of single Ig domains. (**A**) Sequence/topology maps: C1-set, C2-set, FN3, I-set, VNAR, and V-set for CD8a/CD8b and for CD4. See text for details on differences. Capital letters are used for structurally aligned residues. Colors are the same for matching strands in protodomains: blue for strands A and D, green for B and E (a full sheet ABED will read blue–green–green–blue), yellow for strands C and F, orange for strand C’ and G, and red for C” “IgV “linkers”. Colors green and yellow of central strands highlight the self-assembly of strand B-E and C-F of protodomains. FN3 and IgC2 have a similar topology, but FN3 does not share the CCW(L) present in all the selected Ig domains highlighted in central strands. Some IgV domains, such as CD2, however, can also miss that pattern (not shown). IgVs have a C” strand highlighted in red. We have highlighted CD8a and CD8b that form either homo- or heterodimers, with a split A/A’ strand where A’ swaps from one sheet, i.e., antiparallel to strand B and parallel to strand G, while CD4 IgV represents another type where the strand A is only on one sheet in parallel to strand G. VNAR shares the split strand A/A’ of IgVs, which is also present in the Ig-I set. The I-set is very similar to VNARs that have only a few more residues in their HV2 region that encompass a short C’ strand antiparallel to C, and a C’-D linker that is structurally superimposable in VNAR and I-set. In fact, the overall structural alignment is very good with an RMSD of 1.83A and a surprisingly good sequence match (https://structure.ncbi.nlm.nih.gov/icn3d/share.html?c8F4iowua2TAD6ej8, accessed on 27 August 2021) (**B**) Alignment of protodomains in C1-, C2- and I-set domains. Colors, as in A, show the corresponding strands aligning 3 out of 4 strands due to symmetry breaking (see text). In the I-set, the G strand is shown in orange, as well as that which would be C’ when comparing with IgVs, if it were not a protodomain “linker” in the I-set context. RMSD for protodomain alignments are for C1: 1.19 A, FN3: 1.8 A, C2: 1.98 A, I: 2.46 A (iCn3d self-alignment of 2DM3: https://structure.ncbi.nlm.nih.gov/icn3d/share.html?tBzknU8spazpwCup7, accessed on 27 August 2021) (**C**) Alignment of protodomains in IgVs and VNAR domains. The symmetry breaking (see text) also extends to CD4-like IgVs, where strand A is swapped in protodomain 1 vs. strand D in protodomain 2. In other IgVs, such as CD8 or antibody VH-VL domains that we added for completeness, or in VNAR, the strand A is split A/A’, as in the I-set, and keeps a symmetric correspondence with strand D, overall, in all 4 strands AB-CC’ vs. DE-FG, with only a partial symmetry breaking through strand A’. Protodomain alignments consistently show a structural match for conserved and co-evolved residues C in strand B and strand F vs. residues W and (most often) L in strand C and strand E, respectively. RMSD for alignments: for CD8a-IgVs, 1.61 A (CD8b sequences are indicated sequence by homology with mouse CD8b structure (2ATP) whose protodomains superimpose with and RMSD of 0.895 A and 1.54 A), followed by VH-VL protodomains for reference with an RMSD of 1.8, 0.882, and 1.74 A respectively; CD4-IgV protodomains, 1.84 A; VNAR protodomains, 2.31 A.

**Figure 3 biomolecules-11-01290-f003:**
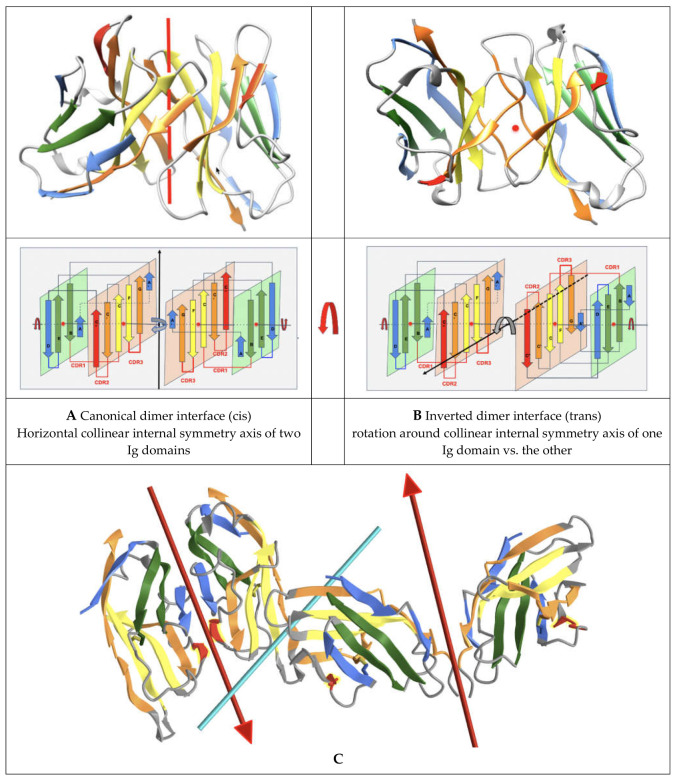
Parallel and antiparallel Ig domain dimerization interfaces. (**A**) Canonical dimer interface using the GFCC’ sheet: The observed quaternary axis of symmetry is vertical. This is the classical interface of antibody VH-VL domains with the CDRs on the same side. It is also found in the Bence Jones protein as a VL-VL or CD8aa as a homodimer and CD8ab as a heterodimer. The figure shows the structure of CD8aa interacting through the GFCC’ sheet in parallel. Its topology/sequence map can be seen in Figure 1 parallel homodimer (PDBid: 1CD8) (https://structure.ncbi.nlm.nih.gov/icn3d/share.html?JcP3sd1gGfqXEBEM8, accessed on 27 August 2021) (**B**) Inverted dimer interface using the GFCC’ sheet: The observed quaternary axis of symmetry is horizontal coming out of the page plane. It corresponds to a rotation around the collinear internal symmetry axis (horizontal left to right) of one Ig domain vs. the other. In this case, CDRs are on opposite sides, the C terminus G strands are pointed in opposite directions. The figure shows the structure of a VL-VL dimer interacting through the GFCC’ sheet in antiparallel to an iCn3D inverted homodimer (PDBID: 7JO8) (https://structure.ncbi.nlm.nih.gov/icn3d/share.html?eGmk1SeSjLyok3p37, accessed on 27 August 2021). As can be observed in the top drawings and in the 3D models (links), the domains are tilted relative to each other [25], this is not represented in the idealized models in the lower drawings. If one considers the tertiary C2 pseudosymmetry of individual domains, when they collinearize, the axes of symmetry cross the quaternary symmetry axis at the center of symmetry and form a higher pseudosymmetry group D2. In most complexes, however, the tertiary and quaternary axes do not cross exactly at the center, as in PD-1/PDL-1. This is an idealized model, naturally, yet in homodimers we observe a quasi D2 pseudosymmetry. The blue–green–yellow–orange coloring associated with each of the individual strands of protodomains A B-C C’ and D E-F G self-assemble as an Ig domain through their central green strands B|E and yellow strands C|F. Symmetry breaking occurs on domain edges through the C” strand in red and through the A-A’ split strand. iCn3D links for (i) a canonical interface of the Bence Jones protein (1REI VL-VL) (https://structure.ncbi.nlm.nih.gov/icn3d/share.html?RK2mWw4w3BfULb2H6, accessed on 27 August 2021); (ii) an inverted interface (7JO8 VL-VL) (https://structure.ncbi.nlm.nih.gov/icn3d/share.html?eGmk1SeSjLyok3p37, accessed on 27 August 2021) and (iii) a comparison of the two with their respective symmetry axes and dimer interactions. (https://structure.ncbi.nlm.nih.gov/icn3d/share.html?EWFzw5ZznZ8ZRsRV9, accessed on 27 August 2021). (**C**) CD86 dimer and CTLA-4 dimer interacting on opposite cell membrane surfaces using a variety of interfaces: CD86 homodimerizes in cis using its strands C”D; CTLA-4 dimerizes in cis (similarly to CD28) on the opposite T-cell surface using strands A’ and G and, most importantly, the hinge connecting the G strand to the transmembrane domain (forming a dimeric cystine bridge not shown in the figure but present in another 3D structure (PDBid:3OSK) https://structure.ncbi.nlm.nih.gov/icn3d/share.html?tzWfkz7TyCrmRsJm7, accessed on 27 August 2021), leaving in both CD86 and CTLA-4 their GFCC’ sheets to interact in trans (PDBid:1I85) (https://structure.ncbi.nlm.nih.gov/icn3d/share.html?ScHMZf2B5i1xQ3JZA, accessed on 27 August 2021).

**Figure 4 biomolecules-11-01290-f004:**
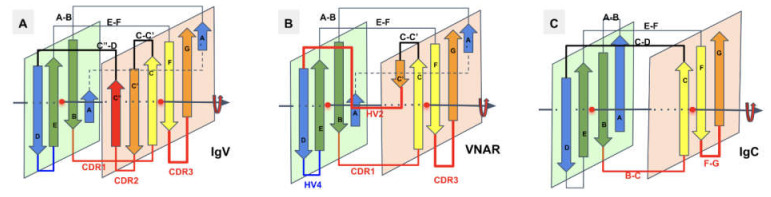
Ig domain topologies for IgV, Shark VNAR, and IgC. (**A**) In IgV domains, the A strand has a flexible hinge in the middle, usually a cis-proline or a stretch of glycines and swaps the upper part of the strand (noted A’) to Sheet B. (**B**) VNAR shows that same domain level organization with two protodomains, yet a much smaller inter-protodomain linker, eliminating the linker’s secondary structure as present in IgV domains: the C” strand and the CDR2 loop. Instead, a short HV2 is observed. In the literature, the C’ strand is usually included in the HV2 region, as it is extremely short. In addition, a hydrophilic set of residues facing out on Sheet B (GF|CC’), rather than being hydrophobic in IgV, do not permit the formation of a symmetric dimer. Dimerization in crystal structures can be observed through HV2, yet they may not be relevant biologically. (**C**) IgC. Here, we consider only the IG C1 set, i.e., the antibody-constant domain-like set, to exemplify an additional protodomain connectivity. In this case, the final domain is formed by a full 4-stranded AB|ED (Sheet A) vs a 3-stranded GFC Sheet B, and no C’ strand. Interestingly, this enables the domain-level dimerization through that 4-stranded Sheet A to form an 8-stranded barrel, as opposed to the IgV that uses the opposite Sheet B to form an 8-stranded dimer barrel.

**Figure 5 biomolecules-11-01290-f005:**
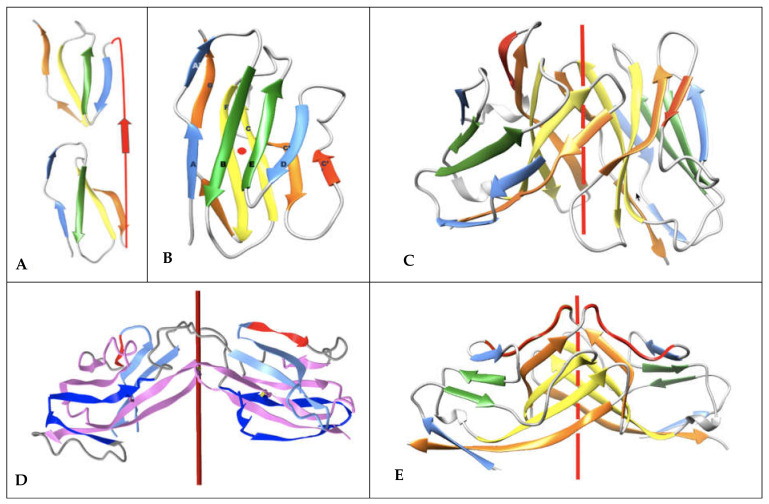
Single IgV and double IgV domain deconstruction. IgV dimers vs. double Ig domain (CD19). (**A**) Single IgV domain schematic structural deconstruction of a (**B**) IgV domain (PDBid:1CD8): Two protodomains AB-CC’ and DE-FG are fused in one domain in antiparallel, i.e., an “inverted topology” in membrane protein terminology. See Figure 1 for details. (**C**) An IgV (canonical) CD8 dimer (https://structure.ncbi.nlm.nih.gov/icn3d/share.html?JcP3sd1gGfqXEBEM8, accessed on 27 August 2021). Homo- or heterodimerization of CD8 occurs through the formation of an 8-stranded central barrel from monomers’ GF|CC’ contacting sheets. Many Ig-based surface receptors use that interface, and so do antibodies in pairing VH-VL domains. (**D**) A swapped “tertiary dimer” (MXRA8; PDBid: 6JO8) https://structure.ncbi.nlm.nih.gov/icn3d/share.html?T53nATsrKyWsDZwX8, accessed on 27 August 2021)—swapping the AB substructure, resulting in a pseudosymmetric tertiary structure in a head to head, resembling a quaternary swapped dimer as in (**F**) but without forming a central barrel; see text for details. Coloring AB dark blue CC’ light blue, C” red, protodomain DE-FG magenta. (**E**) A swapped CD2 IgV quaternary dimer (PDBid:1CDC) (https://structure.ncbi.nlm.nih.gov/icn3d/share.html?4VwaDvKUEMipufmg8, accessed on 27 August 2021) is observed between two IgV domains of CD2 that swap their second respective protodomains DE-FG to lead to a dimer where the first domain is composed of protodomains 1 (AB-CC’) and 4 (DE-FG) and the second Ig domain is composed of protodomains 3 (DE-FG) and 2 (AB-CC’). The linkers (in red) between protodomains 1–2 and 3–4 extend to bridge the two swapped IgV domains. C2 symmetry is preserved. (**F**) Double Ig domain CD19 schematic structural deconstruction (in a CD19 domain the A chain is not present, only A’). (**G**) CD19 double Ig domain (PDBid: 6AL5) (https://structure.ncbi.nlm.nih.gov/icn3d/share.html?LGcQe5UM4nFx7dnL8, accessed on 27 August 2021): the first two protodomains {AB-CC’-DE-FG}_1_ are chained together in a “parallel topology” (with a short linker C’-D in red). The second protodomain pair {AB-CC’-DE-FG}_2_ assembles with the first pair pseudosymmetrically through an “inverted topology”, thanks to a small, intercalated domain linker (in gray). Local pseudosymmetries between protodomains 1 + 4 and 2 + 3 are conserved, as occurs for a regular IgV. The central axis of symmetry of the double Ig and the two local axes of symmetry of the composite Ig domains are shown as red dots perpendicular to the plane of the paper. Color shows local self-associations, at the strand level: as in regular or swapped Igs through their green strands B|E and and yellow strands C|F, and, in addition, the double Ig assembles through the central strands D|D in blue and C’|C’ in orange. The A’ strands in both Igs are associated with the GFCC’ sheet (no A strand).

**Figure 6 biomolecules-11-01290-f006:**
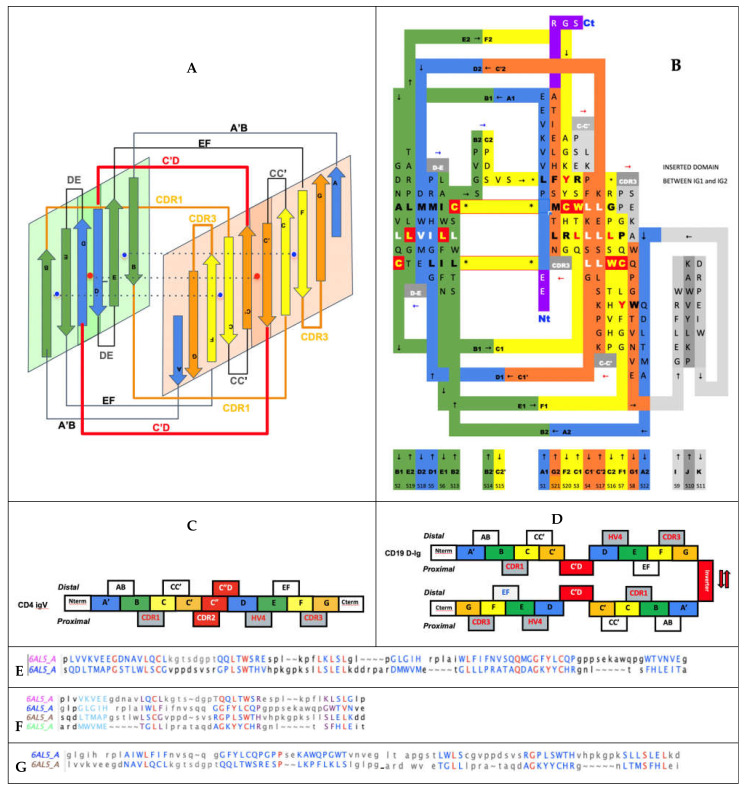
Double IgV domain topology (CD19). (**A**) CD19 double Ig domain topology: Topology diagram with the two sheets facing each other. The central axis of symmetry (red dot) and the two local ones (blue dots) are shown in dashed lines. (**B**) The topology/sequence map showing the pairing of individual residues in complementary beta strands. The comparison with the topology/sequence map for a single Ig domain in Figure 1D shows a conserved organization of composite Ig domains within the double domain, including the CCWL pattern (highlighted in red), as it is in a swapped dimer in Figure 5C. (**C**) Sequence of an IgV with an A’ and no A strand as in CD4, marking the sequence stretches for strands and loops. The (CDR2-C”-C”D loop) linker inverts protodomain 2 A’GFCC’ vs protodomain 1 BED. The CDR loops CDR1, CDR2, and CDR3 and HV4 are all on the same side, here defined as proximal for comparison. (**D**) The CD19 sequence composed of protodomain 1 AB-CC’ and protodomain 2 DE-FG in parallel with a C’D short linker. In the first, a sequential parallel Ig domain CDR1 on the proximal side vs. CDR3 and HV4 on the distal side. The second duplicated parallel Ig domain composed of protodomain 3 and 4 is inverted through a small, inserted domain (in grey in (**B**)). This positions the two parallel Igs to intercalate and form a double Ig domain formed by two long sheets: (BED)1(DEB)2 vs. (A’|GFCC’)1(C’CFG|A’)2. In doing so, two fused composite Ig domains are formed combining p1 + p4 as an Ig domain of topology (AB-CC’|DE-FG) and p2 + p3 as an inverted Ig domain with the topology (DE-FG|AB-CC’), since, in sequence, p2 (DE-FG) precedes p3 (AB-CC). The variable regions (CDR1, CDR3, and HV4) for the first composite Ig domain (p1 + p4) are on the proximal side, as is the case for a regular IgV domain in (**C**), while in the second inverted Ig domain (p3 + p4), variable regions are on the distal side. (**E**) Structural alignment of the two parallel Ig domains within CD19, i.e., p1 + p2 vs. p3 + p4 matching within 1.65 A RMSD. (**F)** Alignment of the 4 individual Ig protodomains: RMSD p1 vs. p2: 1.99 A, p3: 1.29 A, p4: 1.60 A on the 3 last strands BCC’ vs. EFG (since A’ strands are not structurally matched to D strands—see symmetry breaking discussion; they are indicated in light blue). (**G**) Alignment of the two structural Ig domains within the double Ig (CD19). The first p1 + p3 matches the second inverted Ig p2 + p4 within 2.56A RMSD.

**Figure 7 biomolecules-11-01290-f007:**
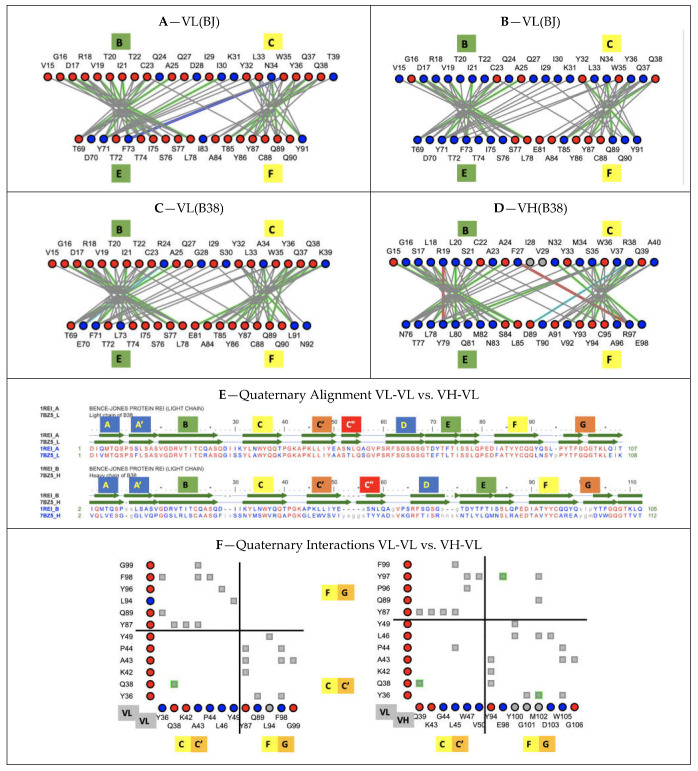
Interaction networks at the Ig protodomain core interface and quaternary Ig-Ig interface. Comparing protodomain interface BC/EF for (**A**) B38 Antibody VL domain (7BZ5) (**B**) B38 Antibody VH domain (**C**) Bence Jones (BJ) VL domain chainA (1REI) (**D**) Bence Jones (BJ) VL domain chainB (https://structure.ncbi.nlm.nih.gov/icn3d/share.html?i2HiYhm3e4P8ecby6, accessed on 27 August 2021)—residue node colors: red = conserved/blue = not conserved between antibody VH-VL B38 and Bence Jones protein dimer VL-VL. Line colors: interactions: grey = VdW contact; green = H-bond; cyan = charge–charge; red = charge–aromatic; blue = charge–aromatic. Cys bonds shown in grey. The interface exhibits a conserved pattern, including, in particular, the CCW residues. (**E**) Quaternary Alignment BJ VL-VL vs. B38 VH-VL: the VH-VL dimer (7BZ5) aligns with a VL-VL dimer (1REI) within 1.53 A RMSD over 202 residues with 57% sequence identity. (**F**) Quaternary interface of BJ homodimer VL-VL (left) vs. B38 heterodimer VH-VL (right) with contributions from protodomain 1 (CC’) and protodomain 2 (FG), forming the GFCC’ sheet (https://structure.ncbi.nlm.nih.gov/icn3d/share.html?aYHnSxgAD6kSoQqn8, accessed on 27 August 2021)—colors as in (**E**)—notice the symmetry in contacts in VL-VL (apart from a small experimental difference) vs. asymmetry in VH-VL, with a highly conserved Q38 vs. Q38/39 H-bonded interaction (green).

## Data Availability

All the structural data used in this study is publicly available through the NCBI web site (https://www.ncbi.nlm.nih.gov/structure/, accessed on 27 August 2021) and the PDB web site (https://www.rcsb.org/, accessed on 27 August 2021). All the analyses, computed data, (such as molecular interactions) and associated molecular visualizations and annotations are available through the iCn3D links provided all along the paper.

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
