# Peer review of "Topological and Structural Plasticity of the Single Ig Fold and the Double Ig Fold Present in CD19"

_biomolecules, 2021, doi:10.3390/biom11091290_

Round 1

Reviewer 1 Report

The author discusses a very interesting and highly relevant topic in this manuscript - Ig-Ig interfaces, their evolution, and their symmetries. 

This is a very timely article, as the understanding of different Ig folds is a critical aspect in protein engineering and design. 

We noticed a typo in line 198 (parralel instead of parallel) and in line 440 there is an additional e in the sentence that might not belong there. 

We ask the author to briefly discuss the dynamics/flexibilities of these interfaces in the discussion (what has already been shown for the VH/VL/ CH1-CL domains, single domain antibodies like VNARs) with MD simulations to complement the manuscript.

Reviewer 2 Report

1.The paper 'Topological and structural plasticity of the single Ig fold and the double Ig fold present in CD19' is a review that needs to be more structured and more concise. Repetitions, redundancies, and unsupported conclusions should be avoided. 

2. IMGT advances in the standardized description of the sequences and structures of the variable and constant domains of the IG, TR, and other IgSF should be taken into account.

2.1. For an easier comparison in the tool, the orientation of a basic V domain of IG and TR and a V-like domain would gain to be presented with the BC, C'C", FG loops at the top, and similarly that of a C domain with the BC and FG loops at the top.

2.2. CDR should be used only for the variable domains of the IG and TR.

2.3. It is rather surprising that there is no mention of the IMGT Collier de Perles on two layers with hydrogen bonds (available in IMGT/3Dstructure-DB, http://www.imgt.org) which bridge sequences and structures. https://www.frontiersin.org/articles/10.3389/fimmu.2014.00022/full

2.4. What not use the IMGT numbering as an option to the PDB entry numbering in the representations? 

In conclusion:

This paper is a confusing draft (too many directions, and everything has already been published by the author on the protodomain and for CD19 well described in the original publication).

The tool is a promising prototype for describing the two layers of a domain (please forget about protodomain here!). It is much better to spend a few weeks/months of work and propose a standardized tool describing the two layers (what it already does) using the IMGT unique numbering. This will be very useful to fill the gap between the information given by the IMGT Collier de Perles on two layers and the PDB 3D structures.

Reviewer 3 Report

The paper “Topological and Structural Plasticity of the single Ig fold and the double Ig fold present in CD19” by Philippe Youkharibache tries to dissect the structural organization of Ig fold using the concept of protodomain. The latter is considered as a sort of “building block” of Ig that by duplication has given rise to the pseudo-symmetrical tertiary Ig domain and subsequently to the quaternary Ig structure.

The paper is highly speculative, but the ideas are presented interestingly and worth being published.

The idea that the overall Ig derives from the duplication of a proto-domain is fascinating, but it is not simple to be demonstrated. If I understand correctly, the pseudo symmetry is limited to the folding, whilst there is not conservation of the amino acid sequence (or, if there is, the a.a. positions in space do not respect the symmetry itself).  

What is not clear to me is to which level this idea is not only speculation, but it can bring to a factual interpretation. It is evident that there is a pseudo-symmetry between the two halves of the Ig, but it could be simply due to convergent evolution.

Is it possible to quantify how much “pseudo” is pseudo? For example, it should be interesting to have some number about the level of this pseudo-symmetry, superimposing the Calpha atoms of the backbone of the two halves and calculate the root mean square deviation about them, comparing this number for different Ig-classes. Are they significant?

Or is the pseudo-symmetry only limited to the overall topology, i.e. as appears in the figures?

In addition, one has to bear in mind that the interaction surface between two proteins is determined by the residues present, and not only by the overall fold. The same fold with a different amino acid sequence could give rise to a fully different surface of interaction. Does the portion of the surface determined by one half (one protodomain) correspond to that of the other?

In the conclusions, the author states that this idea could open the way for new technological improvements, for example in the design of new nanobodies. Could they figure out how this could be done?  

Minor points:

Fig. 4 is mentioned just after Fig. 1

Round 2

Reviewer 2 Report

I thank the author for answering my comments. I understand that in a short laps of time, deep changes cannot be done.

However standards developped in immunoinformatics should be used for topology: 1. the IMGT unique numbering for V or C domain which is valid for any V-like or C-like domain of the IgSF, 2. the classical orientation (loop at the top) and label (CDR-IMGT only for IG and TR).  

Author Response

point 1:  standards developped in immunoinformatics should be used for topology: 1. the IMGT unique numbering for V or C domain which is valid for any V-like or C-like domain of the IgSF, 2. the classical orientation (loop at the top) and label (CDR-IMGT only for IG and TR).  

response 1: 1. I have added IMGT numbering for reference in a supplement figure, along  with Kabat's.  2. I cannot change the paper graphical representations at this stage without a lot of tedious error prone manipulations.  We may be one year away at best from having a tool to produce graphical representations, at that stage we will be able to enable changes in orientation.  I understand your concern for IMGT’s standard to be used, but there is no standard used at this stage by crystallographers on one end or in by our clinicians and biologists developing antibodies, who may use Kabat’s numbering along the sequence, while none of them use any graphical topological representation.  I suggested that you to reach out to NCBI for adopting a standard first in numbering and then ideally of course for graphical 2D topological representation, which I would support.

Let me share with you an NCBI CDD curator's comment to me “For the past 20 years, all published papers were incorrectly describing CD19 as IgC2-type, which categorizes Ig domains without Strand D, although CD19 indeed do contain Strand D.  You have to emphasize your point that your paper is the first paper ever describing the domain architecture of CD19 correctly.”

Reviewer 3 Report

In my opinion, the paper can be published in his current form

Author Response

I thank the reviewer for constructive feedback and comments that allowed me to clarify protodomains interactions in particular.